# LLM4GCL: CAN LARGE LANGUAGE MODEL EMPOWER GRAPH CONTRASTIVE LEARNING?

## ABSTRACT

Graph contrastive learning (GCL) has made significant strides in pre-training graph neural networks (GNNs) without requiring human annotations. Previous GCL efforts have primarily concentrated on augmenting graphs, assuming the node features are pre-embedded. However, many real-world graphs contain textual node attributes (e.g., citation network), known as text-attributed graphs (TAGs). The existing GCL methods often simply convert the textual attributes into numerical features using shallow or heuristic methods like skip-gram and bag-of-words, which cannot capture the semantic nuances and general knowledge embedded in natural language. Motivated by the exceptional capabilities of large language models (LLMs), like ChatGPT, in comprehending text, in this work, we delve into the realm of GCL on TAGs in the era of LLMs, which we term LLM4GCL. We explore two potential pipelines: *LLM-as-GraphAugmentor* and *LLM-as-TextEncoder*. The former aims to directly leverage LLMs to conduct augmentations at the feature and structure levels through prompts. The latter attempts to employ LLMs to encode nodes' textual attributes into embedding vectors. Building on these two pipelines, we conduct comprehensive and systematic studies on six benchmark datasets, exploring various feasible designs. The results show the promise of LLM4GCL in enhancing the performance of state-of-the-art GCL methods. Our code and dataset will be publicly released upon acceptance.

## 1 INTRODUCTION

Graph contrastive learning (GCL) has demonstrated remarkable efficacy in the pre-training of graph neural networks (GNNs) using unlabeled data (Wu et al., 2020). Existing GCL research, exemplified by GraphCL (You et al., 2020) and BGRL (Thakoor et al., 2021), operate by creating two augmented views of the input graph and subsequently training GNN encoder to produce similar representations for both views of the same node, as shown in Figure 1. By pre-training GNN models using a large number of unlabeled graphs, the pre-trained model or learned representations can be employed to enhance various downstream tasks such as link prediction (Zhang & Chen, 2018; Pan et al., 2018), node classification (Hassani & Khasahmadi, 2020; Zhu et al., 2020a; Qiu et al., 2020; Gong et al., 2023), and graph classification (Xu et al., 2021; Suresh et al., 2021; Xia et al., 2022).

Despite the plethora of GCL methods proposed in recent years (Veličković et al., 2018b; Zhu et al., 2021b; Hu et al., 2020b; Zhang et al., 2021a; Bielak et al., 2022), they exhibit limitations when confronted with graphs that possess rich textual descriptions, often referred to as text-attributed graphs (TAGs). A typical example of TAGs is citation network, where each node represents a research paper, accompanied by node attributes in the form of titles and abstracts. These textual descriptions serve as a valuable source of information for enhancing graph representation learning, as text is highly expressive, capturing intricate semantic nuances. However, as illustrated in Figure 1, previous GCL efforts simply employ textual attributes to derive numerical features using shallow embedding models like Word2vec (Mikolov et al., 2013) or Bag-of-Words (BoW) (Harris, 1954). These shallow embeddings are suboptimal, since they cannot capture the complexity of semantic features (He et al., 2023; Chen et al., 2023). Moreover, they conduct feature and structure augmentation in an attribute-agnostic manner, relying solely on stochastic perturbation functions like feature and edge masking; that is, the valuable textual attributes have not been fully leveraged in graph augmentation.

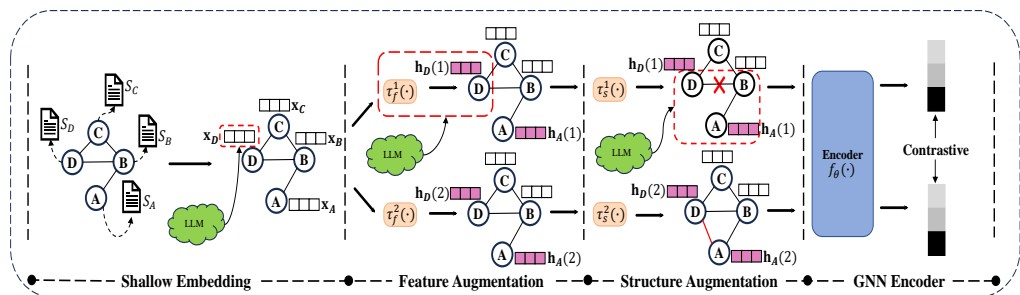

Figure 1: The learning paradigm of GCL methods on TAGs. Given a TAG $\mathcal{G} = \{\mathcal{V}, \mathcal{S}, \mathbf{A}\}$, previous GCL endeavors first adopt shallow embedding functions to convert the textual attribute ($\mathcal{S}$) into numerical features ($\mathbf{X}$). After that, the feature-level and structure perturbation functions (i.e., $\tau_f$ and $\tau_s$) will be subsequently applied to the node features $\mathbf{X}$ and graph structure $\mathbf{A}$ to generate two augmented graphs for contrastive learning. LLMs have the potential to advance graph augmentation, including feature and structure aspects, and the embedding process.

Motivated by the formidable capabilities of large language models (LLMs) [1], such as Chat-GPT (Brown et al., 2020), there has been a growing focus on harnessing LLMs for graph representation learning on TAGs (Zhang, 2023; Yang et al., 2021; Ye et al., 2023). For instance, GLEM (Zhao et al., 2022) proposes a method involving the fine-tuning of a pre-trained language model (PLM), like BERT (Devlin et al., 2018) and DeBERTa (He et al., 2020), to predict pseudo-labels generated by a GNN model. TAPE (He et al., 2023) takes an approach where ChatGPT is initially employed to generate new textual features for each graph node, followed by supervised fine-tuning of DeBERTa using these newly generated texts. SimTeG (Duan et al., 2023) further incorporates low-rank adaptation (Hu et al., 2021a) into the fine-tuning process of the PLM model. However, all the methods mentioned above are designed for fine-tuning PLMs on graphs in the *supervised* setting. The problem of how to leverage LLMs for textual graphs in the *self-supervised* setting remains unexplored.

To bridge the gap, we present LLM4GCL, the first comprehensive and systematic study on harnessing LLMs for GCL. Our study is designed to delve into the following key research questions. *i)* How can we utilize LLMs for enhancing graph augmentations at both the feature and structural levels? *ii)* How can we fine-tune a pre-trained PLM in an unsupervised manner to enhance its capacity for encoding textual node attributes and structural relationships?

**Contributions.** We outline our main contributions below.

1. We explore two potential LLM4GCL pipelines designed to enhance contrastive learning on textual graphs: *LLM-as-GraphAugmentor* and *LLM-as-TextEncoder*. The former seeks to employ LLM (e.g., ChatGPT) for perturbing the original graph, focusing on both feature and structural aspects, thereby enhancing conventional heuristic augmentation strategies. The latter involves fine-tuning a PLM using self-generated signals, followed by the utilization of the resulting node features in standard GCL methods.

2. For *LLM-as-GraphAugmentor*, we utilize LLM to directly execute feature and structural augmentations through various prompt templates. **Our key observations**: ❶ Instead of using the original textual node attributes, LLM can generate more informative textual descriptions with appropriate prompts. However, integrating structure information into this process poses a non-trivial challenge. ❷ In comparison to standard structural augmentation methods like edge masking, LLM demonstrates the potential to generate highly competitive graph structures for GCL. ❸ It becomes feasible to directly utilize the generated textual attributes and graph structure to replace one augmented view, indicating the potential of LLM in simplifying the graph augmentation process.

3. Regarding *LLM-as-TextEncoder*, we investigate several self-supervised fine-tuning strategies to adapt general PLMs to the graph domain for more effective encoding of textual attributes. **Our key findings:** ❹ A PLM model cannot be directly applied to encode textual node attributes effec-

---

[1]In this work, "LLM" refers to large language models, such as ChatGPT, which require powerful computational devices, while "PLM" denotes small language models fine-tunable on common GPUs, e.g., BERT.

tively. In some cases, even shallow embedding techniques like BoW outperform it. ❺ Incorporating structural information during fine-tuning can yield benefits compared to the standard masked language modeling strategy.

## 2 PRELIMINARY

In this section, we introduce notations, formalize the research problem of this work, and illustrate prospective opportunities for harnessing language models to enhance GCL on TAGs.

***Text-Attributed Graphs.*** We are given a TAG $\mathcal{G} = \{\mathcal{V}, \mathcal{S}, \mathbf{A}\}$ with $N$ nodes, where $\mathcal{V}$ denotes the node set, and $\mathbf{A} \in \mathbb{R}^{N \times N}$ represents the adjacency matrix. For each node $v \in \mathcal{V}$ is associated with a textual attribute $S_v$, and $\mathcal{S} = \{S_v | v \in \mathcal{V}\}$ is the attribute set.

In this work, we study self-supervised learning on TAGs. Specifically, the goal is to pre-train a mapping function $f_\theta : \mathcal{S} \times \mathbf{A} \to \mathbb{R}^D$, so that the semantic information in $S$ and the topological structure in $\mathbf{A}$ could be effectively captured in the $D$-dimensional space in a self-supervised manner.

***Graph Neural Networks.*** For graph-structure data, graph neural networks (GNNs) are often applied to instantiate $f_\theta$. Specifically, the goal of GNNs is to update node representation by aggregating messages from its neighbors, expressed as:

$$\mathbf{h}_v^{(k)} = \text{COM}(\mathbf{h}_v^{(k-1)}, \text{AGG}(\{\mathbf{h}_u^{(k-1)} : u \in \mathcal{N}_v\})), \tag{1}$$

where $\mathbf{h}_v^{(k)}$ denotes the representation of node $v$ at the $k$-th layer and $\mathcal{N}_v = \{u | \mathbf{A}_{v,u} = 1\}$ is a direct neighbor set of $v$. In particular, we have $\mathbf{h}_v^{(0)} = \mathbf{x}_v$, in which $\mathbf{x}_v = \text{Emb}(S_v) \in \mathbb{R}^F$ is a $F$-dimensional numerical vector extracted from $v$'s textual attribute $S_v$ and $\text{Emb}(\cdot)$ stands for embedding function. The function AGG is used to aggregate features from neighbors (Kipf & Welling, 2016), and function COM is used to combine the aggregated neighbor information and its own node embedding from the previous layer (Vaswani et al., 2017).

***Graph Contrastive Learning on TAGs.*** Let $\tau_f : \mathbb{R}^F \to \mathbb{R}^F$ and $\tau_s : \mathcal{V} \times \mathcal{V} \to \mathcal{V} \times \mathcal{V}$ represent the feature-level and structure-level perturbation functions, respectively. An example of $\tau_f$ is feature masking (Jin et al., 2020), while for $\tau_s$, edge masking (Zhu et al., 2021a) serves as a typical illustration. Previous GCL endeavors (You et al., 2021; Yin et al., 2022; Zhang et al., 2022) typically generates two augmented graphs, $\mathcal{G}_1 = (\mathbf{A}_1, \mathbf{X}_1)$ and $\mathcal{G}_2 = (\mathbf{A}_2, \mathbf{X}_2)$, utilizing perturbation functions. Here, $\mathbf{X}_1 = \{\tau_f^1(\mathbf{x}_v) | v \in \mathcal{V}\}$, $\mathbf{A}_1 = \tau_s^1(\mathbf{A})$, $\mathbf{X}_2 = \{\tau_f^2(\mathbf{x}_v) | v \in \mathcal{V}\}$, and $\mathbf{A}_2 = \tau_s^2(\mathbf{A})$. Subsequently, two sets of node representations are acquired for the two views using a shared GNN encoder, denoted as $\mathbf{H}_1$ and $\mathbf{H}_2$, respectively. Finally, the GNN encoder is trained to maximize the similarity between $\mathbf{H}_1$ and $\mathbf{H}_2$ on a node-wise basis. In this study, we primary focus on three state-of-the-art methods, namely GraphCL (You et al., 2020), BGRL (Thakoor et al., 2021), and GBT (Bielak et al., 2022), for experimentation.

## 3 LLM4GCL

Figure 1 illustrates the learning paradigm of standard GCL methods. While effective, these methods have limitations in harnessing informative textual attributes because shallow models often struggle to capture intricate semantic features. As depicted in Figure 1, we investigate two potential avenues for incorporating LLMs into GCL, focusing on graph augmentation and feature extraction. This section will introduce detailed strategies for utilizing LLMs for feature/structure augmentation (Section 3.1) and fine-tuning a PLM to improve text embedding (Section 3.2).

### 3.1 LLM AS GRAPH AUGMENTOR

Graph augmentation plays a pivotal role in the success of GNN pre-training. A good augmentation scheme is expected to retain the original graph's semantic information in a modified version (Lee et al., 2022; Tan et al., 2022). Nonetheless, previous augmentation techniques (e.g., feature or edge masking) may fall short in TAGs, mainly because they primarily revolve around stochastic perturbations (Ding et al., 2022a) within the feature or relational space, neglecting the informative textual semantics. In this subsection, we explore the potential of LLMs (e.g., ChatGPT) for enhancing graph

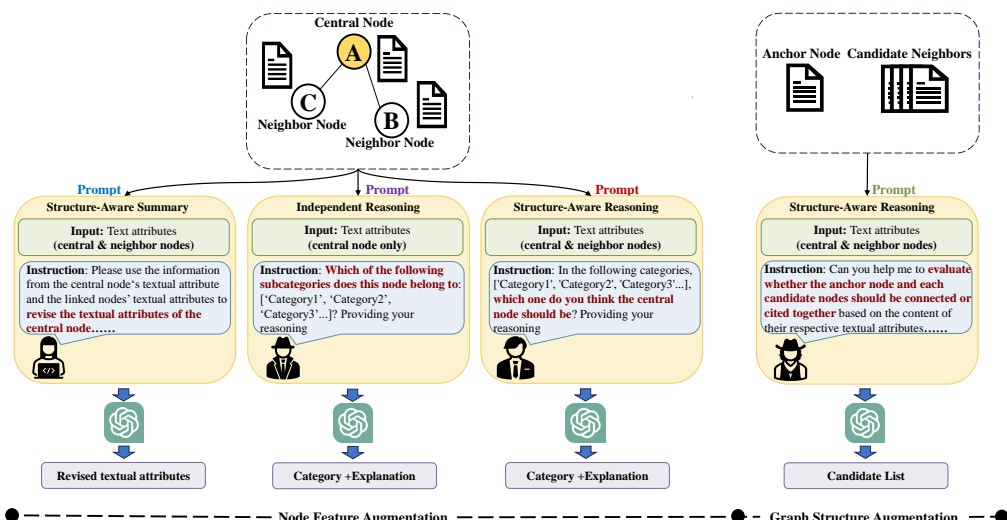

Figure 2: LLM-as-GraphAugmentor. **Left**: LLMs are emloyed to perturb node features by influencing the input textual attributes. **Right**: LLMs are utilized to create new graph structures by modifying and adding edges between nodes.

augmentation, depicted in Figure 2. In Section 3.1.1 and Section 3.1.2, we introduce strategies for performing feature-level and structure-level augmentations using LLMs, respectively.

### 3.1.1 LLM FOR FEATURE AUGMENTATION

Given a node $v$ and its textual attribute $S_v$, traditional GCL methods typically create an augmented feature vector $\hat{\mathbf{x}}_v$ using purely stochastic functions, i.e., $\hat{\mathbf{x}}_v = \tau_f(\mathbf{x}_v) = \tau_f(\text{Emb}(S_v))$. However, this approach only introduces perturbations within the numerical space transformed by the $\text{Emb}(\cdot)$ module, which cannot effectively manipulate the original input textual attribute. To overcome this limitation, we propose to use LLMs to directly perturb the input text $S_v$ and obtain an augmented textual attribute $\hat{S}_v$ through three prompt templates (refer to Figure 2 (left)) outlined below.

**Structure-Aware Summarization (SAS).** Let $\mathcal{S}_v^N = \{S_u | v \in \mathcal{N}_v\}$ represent the textual attribute set of node $v$'s neighbors. The idea of SAS is to query the LLM to create a summary of the anchor node $v$ by comprehending the semantic information from both its neighbors and itself. Specifically, for each node $v$, we construct a prompt that incorporates the textual attributes of the anchor node and its neighbors, denoted as $\{S_v, \mathcal{S}_v^N\}$, along with an instruction for revising its textual attribute. The general prompt format is illustrated in the left panel of Figure 2 (left). Specific prompt templates for various datasets are detailed in Table 10 of Appendix C. Finally, we employ these summarized textual attributes to represent the augmented attribute $\hat{S}_v$.

**Independent Reasoning (IDR).** In contrast to SAS, which concentrates on text summarization, IDR adopts an "open-ended" approach when querying the LLM. This entails instructing the model to make predictions across potential categories and to provide explanations for its decisions. The underlying philosophy here is that such a reasoning task will prompt the LLM to comprehend the semantic significance of the input textual attribute at a higher level, with an emphasis on the most vital and relevant factors (He et al., 2023). Following this principle, for each node $v$, we generate a prompt that takes the textual attribute of the anchor node as input and instructs the LLM to predict the category of this node and provide explanations. The general prompt format is illustrated in the middle panel of Figure 2 (left). Specific prompt templates for different datasets are detailed in Table 9 of Appendix C. We utilize the prediction and explanations to represent the augmented attribute $\hat{S}_v$.

**Structure-Aware Reasoning (SAR).** Taking a step beyond IDR, SAR integrates structural information into the reasoning process. The rationale for this lies in the notion that connected nodes can aid in deducing the topic of the anchor node. Specifically, for each node $v$, we devise a prompt that encompasses the textual attributes of the anchor node $S_v$ and its neighbors $S_v^N$, along with an open-

ended query concerning the potential category of the node. The general prompt format is given in the right panel of Figure 2 (left). More detailed prompt templates for various datasets are provided in Table 11 of Appendix C. Similar to IDR, we employ the prediction and explanations to denote the augmented attribute $\hat{S}_v$.

To reduce the query overhead of ChatGPT, we randomly sample 10 neighbors for each anchor node in structure-aware prompts (i.e., SAS and SAR) in our experiments.

### 3.1.2 LLM for Structure Augmentation

In addition to feature augmentation, structural augmentation is another popular strategy in state-of-the-art GCL methods (Liu et al., 2022). Let $\mathbf{A}$ represent the adjacency matrix of a TAG. Previous efforts have typically relied on structural perturbation functions $\tau_s$, which involve operations like randomly deleting or adding edges, to generate the augmented structure $\hat{\mathbf{A}}$. While this approach has been widely adopted in the literature, its attribute-agnostic nature may not be optimal for capturing the complementary information between attributes and relational data, a phenomenon validated in the research community (Huang et al., 2017). To address this limitation, we propose to leverage LLMs to perturb the graph structure through a graph structure augmentation prompt defined below.

**Graph Structure Augmentation (GSA).** Let $N_v$ and $\bar{N}_v$ denote the connected and disconnected node sets of $v$, respectively. We query the LLM model to predict if nodes in $N_v$ (or $\bar{N}_v$) should be disconnected (or connected) to the anchor node $v$. More precisely, we develop a prompt template, as shown in Figure 2 (right), to answer this question by considering their textual attributes. For instance, given the textual attribute $S_v$ of the anchor node and its textual attribute set of neighbors $S_v^N$, we ask the LLM to decide if each node in $N_v$ should be connected to $v$. The specific prompt templates for different datasets are provided in Table 12 in the Appendix C. Subsequently, we construct an augmented structure $\hat{\mathbf{A}}$ by either dropping or adding edges based on LLM's decisions.

To enhance query efficiency, we initially employ shallow embedding algorithms, such as BoW, to assess pairwise similarity between two nodes. Subsequently, we select the top 20 nearest neighbors from $N_v$ in descending order and the top 20 unconnected nodes from $\bar{N}_v$ in ascending order based on their scores for querying ChatGPT in our experiments. More empirical details can be found at Appendix C.2.

### 3.2 LLM as text encoder

We have demonstrated the use of LLMs for graph augmentations. In Section 3.1, our remaining question is how to implement the embedding function $\mathrm{Emb}(\cdot)$, which is responsible for transforming the (augmented) textual attribute ($S_v$ and $\hat{S}_v$) into an embedding vector ($\mathbf{x}_v$ and $\hat{\mathbf{x}}_v$) using LLMs. One naive approach is to directly employ pre-trained LLMs to encode $S_v$, thereby endowing GCL with the capability to capture richer semantic meanings. However, this direct application of LLMs might yield unsatisfactory performance due to the mismatch between the TAG dataset and general text data. Therefore, fine-tuning procedures are necessary. In this subsection, we present three strategies for fine-tuning LLMs for GCL (See Figure 11 in Appendix C.3) elaborated as follows. It is worth noting that we use PLMs rather than LLMs in the following description due to computational constraints that prohibit fine-tuning LLMs (e.g., ChatGPT) in practice.

**Masked Language Modeling (MLM).** The idea is to directly fine-tune the model on textual attributes. Following (Devlin et al., 2018), our approach entails model fine-tuning through the masking of tokens within the sentence. Specifically, for each textual attribute $S_v = \{w_1, w_2, ..., w_{n_v}\}$ with $n_v$ tokens, where $w_i$ denotes a token, we randomly mask $\omega\%$ of tokens in $S_v$ and replace them with a special MASK token. We represent the set of masked tokens as $S_{\mathrm{mask}}$, while the set of observed tokens as $S_v^{\backslash S_{\mathrm{mask}}}$. The objective of MLM is formally expressed as follows:

$$\mathcal{L}_{\mathrm{MLM}} = \sum_{w_i \in S_{\mathrm{mask}}} \log P_\theta(w_i | S_v^{\backslash S_{\mathrm{mask}}}), \tag{2}$$

where $\theta$ is the model weight of the PLM. In our experiments, we fix the value of $\omega$ at 15.

**Topology-Aware Contrastive Learning (TACL).** One limitation of MLM is that its fine-tuning procedure can not learn the topology information of the graph. To address this, we propose fine-tuning a PLM using contrastive learning, inspired by (You et al., 2020). Given node $v$ and one of its

connected node $u$. Let $\mathbf{x}_v$ and $\mathbf{x}_u$ respectively denote the representations of node $v$ and $u$ generated by the PLM model and $\text{sim}()$ be the cosine similarity function, the training objective of TACL is:

$$\mathcal{L}_{\text{TCL}} = - \sum_{u \in N_v^{\text{pos}}} \log \frac{\exp(\text{sim}(\mathbf{x}_v, \mathbf{x}_u)/\tau)}{\sum_{n=1, n \neq u}^{B} \exp(\text{sim}(\mathbf{x}_v, \mathbf{x}_n)/\tau)}, \tag{3}$$

where $N_v^{\text{pos}}$ comprises $K$ nodes randomly sampled from $N_v$, with $K$ being a hyperparameter. $\tau$ is the temperature parameter, and $B$ denotes the batch size.

**Multi-Scale Neighborhood Prediction (GIANT).** Unlike TACL, we adopt the approach from GIANT (Chien et al., 2022) to reconstruct all neighbors by transforming this task into a multi-label classification problem. However, directly fine-tuning the PLM on a high-dimensional output space of size $|\mathcal{V}|$ is computationally infeasible. To address this challenge, GIANT employs the formalism of extreme multi-label classification (XMC). The key concept is to construct a hierarchical node cluster tree using the balanced k-means algorithm, based on the PIFA features (Zhang et al., 2021b). Subsequently, the PLM is pre-trained to match the most relevant clusters in a top-down manner. For a detailed mathematical exposition, interested readers are encouraged to consult (Chien et al., 2022).

By employing these three fine-tuning strategies, we can efficiently adapt a PLM model to TAGs. Once fine-tuned, this model becomes capable of converting both the original and augmented textual attributes into numerical features.

## 4 EXPERIMENTS

Throughout the experiments, we aim to address the following research questions. **RQ1:** Is GCL helpful on TAGs, especially when compared to standard GNN methods? **RQ2:** Is a general PLM without fine-tuning sufficient to encode textual attributes, and how do the proposed fine-tuning strategies for text encoding perform? **RQ3:** How effective is LLM in generating augmented features and structures? **RQ4:** Which types of language models are best suited for GCL on TAGs? **RQ5:** Can LLM-as-TextEncoder also enhance the performance of generative self-supervised learning methods for graphs? **RQ6:** How do the textual embeddings generated by LLM-as-TextEncoder compare with those generated by shallow methods?

### 4.1 EXPERIMENTAL SETUP

**Datasets.** We evaluate the proposed LLM4GCL framework using five publicly available TAG datasets. These datasets encompass two citation networks, namely PubMed and Ogbn-Arxiv (Arxiv), and three E-commerce datasets extracted from Amazon (Ni et al., 2019), including Electronics-Computers (Compt), Books-History (Hist), and Electronics-Photography (Photo). Unless specified otherwise, we primarily focus on the semi-supervised setting. More details on these datasets and corresponding data splits can be found in Appendix A.

**Baselines.** We consider three type of baseline algorithms for comparison, including two standard GNNs methods GCN (Kipf & Welling, 2016) and GAT (Veličković et al., 2018a), three representative GCL methods GraphCL (You et al., 2020), BGRL (Thakoor et al., 2021), and GBT (Bielak et al., 2022)), and three popular language models BERT (Devlin et al., 2018), DeBERTa (He et al., 2020), and RoBERTa (Liu et al., 2019). Please refer to Appendix B.1 for more experimental details.

**Implementation details.** For the reproducibility of our experiments, we employ GNN implementations from the PyG package (Fey & Lenssen, 2019). For the GraphCL, BGRL, and GBT methods, we closely adhere to the procedures outlined in (Zhu et al., 2021a). We leverage PLMs from Huggingface for feature extraction and utilize OpenAI's ChatGPT 3.5 for graph augmentation. More implementation details are provided in Appendix B.1.

### 4.2 IS GRAPH CONTRASTIVE LEARNING HELPFUL ON TAGS?

Before experimenting with our proposed LLM-based augmentation and text encoding strategies, it is necessary to understand how the existing GCL methods perform on TAGs. To answer **RQ1**, we evaluate the performance of three representative state-of-the-art GCL methods against the standard

| Method | Feat Type | PubMed | Arxiv | Compt | Hist | Photo |
|--------|-----------|--------|-------|-------|------|-------|
| GCN | SE | 77.41±1.22 | 71.74±0.29 | 59.87±0.63 | 62.13±0.26 | 63.69±0.66 |
| GAT | | 79.00± 0.71 | **72.60±0.16** | 55.81±0.49 | 58.61±0.93 | 56.93±0.93 |
| | SE | **79.61±0.13** | 71.82±0.27 | 55.40±0.04 | 69.84±0.42 | 57.98±0.09 |
| | FIX | 68.81±1.25 | 69.37±0.45 | 20.52±1.08 | 67.81±1.11 | 29.94±0.22 |
| BGRL | MLM | 70.19±1.69 | 69.78±0.12 | 21.62±1.08 | 65.41±0.40 | 30.33±0.20 |
| | GIANT | **81.38±0.07** | **73.14±0.21** | 74.23±0.56 | 74.16±0.83 | 71.65±0.61 |
| | TACL | **80.82±0.46** | **73.28±0.15** | **74.85±0.32** | **73.96±0.61** | **73.00±0.43** |
| | SE | 79.44±1.31 | 68.55±0.52 | 69.53±0.26 | 71.62±1.38 | 68.56±0.95 |
| | FIX | 71.96±1.28 | 67.96±0.11 | 51.73±0.08 | 69.81±0.57 | 53.34±0.34 |
| GBT | MLM | 75.26±3.79 | 68.93±0.48 | 55.62±0.37 | 69.57±0.40 | 55.58±0.40 |
| | GIANT | 75.64±2.06 | 62.64±0.22 | **76.87±0.36** | 71.89±0.63 | **74.65±0.69** |
| | TACL | 74.51±2.04 | 71.41±0.55 | **77.07±0.21** | **73.49±0.55** | **75.18±1.08** |
| | SE | 76.48±0.71 | 68.41±0.34 | 51.74±0.75 | 54.21±0.48 | 53.21±0.47 |
| | FIX | 62.98±1.93 | 56.40±0.33 | 34.46±0.80 | 63.63±2.14 | 47.67±0.88 |
| GraphCL | MLM | 65.82±2.17 | 58.04±0.15 | 37.54±1.18 | 63.70±0.30 | 50.37±0.59 |
| | GIANT | 79.13±0.70 | 50.88±0.38 | 74.24±0.24 | 71.14±1.38 | 71.40±0.62 |
| | TACL | 79.15±0.55 | 70.67±0.08 | 71.89±1.54 | **74.39±0.59** | 72.35±0.55 |

Table 1: Accuracies of GCL methods using different input features generated by PLM and three fine-tuning strategies. "SE"/"Fix" indicates features generated by standard shallow models/general PLM models. Highlighted are the top **first**, **second**, and **third** results.

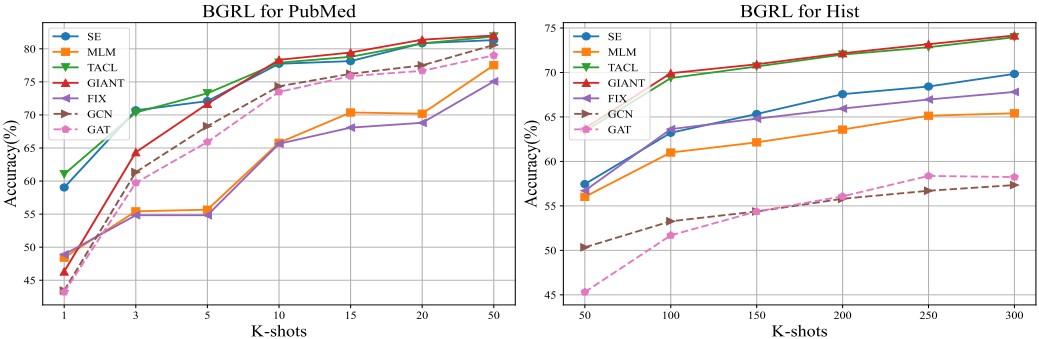

Figure 3: Few-shot learning results on BGRL. Please refer to Appendix B.2.1 for more results.

GNN counterparts in the semi-supervised setting. The results are reported in Table 1. We make the following observations.

① **GCL methods can enhance the performance of GNNs on TAGs in semi-supervised scenarios.** From Table 1, at least one of the GCL methods using the shallow embeddings (i.e., SE) outperforms the standard GNNs across five datasets. Notably, GBT exhibits clear superiority over GCN and GAT on the three E-commerce datasets (Compt, Hist, and Photo), which validates the potential of GCL on TAGs. Nevertheless, ② **GCL methods could degrade the performance in many cases.** Specifically, GraphCL, BGRL, and GBT with shallow embeddings underperform the standard GNNs on 5, 2, and 1 of the datasets, respectively. We conjecture that this stems from the limitations of current GCL methods on TAGs, which do not effectively capture semantic information during augmentation. Furthermore, from Figure 3, ③ **Pre-training GNN models using contrastive learning can improve their performance in few-shot cases.** When the number of available examples is limited, the pre-trained GNN models still achieve highly competitive results. Conversely, GCN and GAT experience a significant performance drop in these few-shot cases. This discrepancy can be attributed to the capability of GCL methods to compel the GNN encoder to acquire more informative and invariant node representations across augmented views (Liu et al., 2021). These findings underscore the importance of exploring contrastive learning on text-attributed graphs.

## 4.3 CAN LLMs ENHANCE THE ENCODING OF TEXTUAL ATTRIBUTES?

To address **RQ2** about the potential of using LLMs to enhance the encoding of textual attributes on TAGs, we investigate the effectiveness of the three fine-tuning strategies proposed in Section 3.2, namely MLM, TACL, and GIANT. For comprehensive details of our experimental setups, please refer to Appendix B.2.2. Table 1 lists the results, where we can make the following key observations.

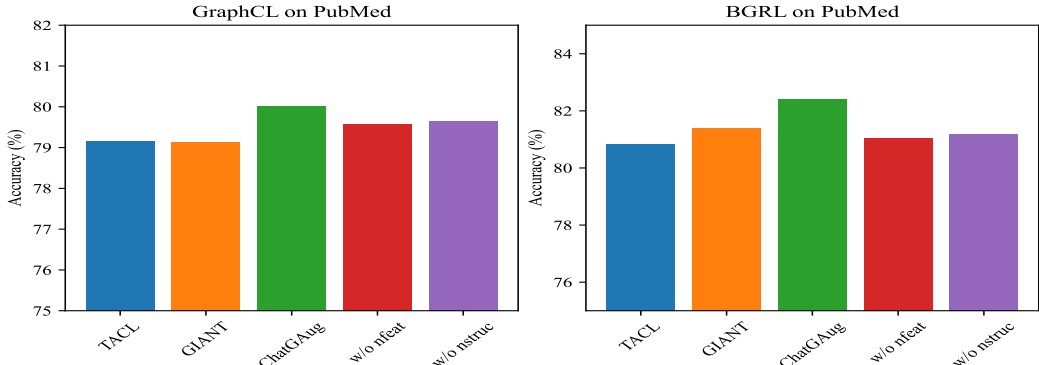

Figure 4: The performance of LLM4GCL, when augmented with both features and structure by ChatGPT, is denoted as ChatGAug. The variants "w/o nfeat" and "w/o nstruc" represent scenarios where augmented features and augmented structure are not considered, respectively.

④ **Directly using the text encoding obtained from a general PLM without fine-tuning does not improve the performance.** From Table 1, we observe that the features generated by general PLM without fine-tuning (FIX in the table) exhibit notably inferior performance compared to the classical features derived from shallow models (SE) across three GCL backbones. This could be explained by the negative transfer issue associated with pre-trained models (Ding et al., 2022b). Meanwhile, ⑤ **the standard masked language modeling strategy is ineffective at fine-tuning PLMs for TAGs.** Although the MLM outperforms FIX by a consideration margin, it still falls short when compared with shallow models across all datasets. This could be attributed to the fact that the MLM overlooks the crucial topological structure. Furthermore, ⑥ **by integrating structure information, both TACL and GIANT can enhance the performance of GCL methods.** Table 1 and Figure 3 show that TACL and GIANT generally outperform both FIX and MLM strategies across all scenarios. Moreover, BGRL, GBT, and GraphCL generally perform better when utilizing node features generated by TACL and GIANT, compared to the results achieved by shallow models.

## 4.4 How effective is LLMs in performing graph augmentations?

While PLMs can enhance textual embeddings, the enhancements are not consistent across all the datasets. In this section, we conduct experiments to assess the feasibility of utilizing LLMs for graph augmentation to achieve further improvements (**RQ3**). Specifically, we investigate the impact of three feature augmentation prompts, namely SAS, IDR, and SAR, as presented in Section 3.1.1, as well as the structure augmentation prompt (i.e., GSA) introduced in Section 3.1.2. The detailed experimental settings are provided in Appendix B.2.3. Figure 4 summarizes the results, from which we draw several key observations.

⑦ **By enhancing standard GCL methods with our augmented features and structures (referred to as "Chat-GAug"), their performance can be further improved.** In Figure 4, we can see that the ChatGAug variant generally outperforms the best baselines mentioned in Table 1, including TACL and GIANT. Similar findings are replicated in other evaluation datasets, as illustrated in Appendix. Furthermore, ⑧ **Leveraging LLMs (e.g., ChatGPT) to jointly augment node features and graph structures is more effective than augmenting them individually.** Figure 4 illustrates that ChatGAug consistently outperforms the "w/o nfeat" and "w/o nstruc" variants in nearly all cases. These comparisons validate that the LLM-based augmentation in the feature and structure levels are complementary.

|  | Method | SAS | IDR | SAR |
|---|---|---|---|---|
| PubMed | GBT | 73.18±2.45 | 66.94±1.23 | 75.12±1.74 |
|  | GraphCL | 79.43±0.74 | 78.10±1.70 | 77.79±1.33 |
|  | BGRL | 80.47±1.14 | 83.01±0.26 | 81.44±0.29 |
| Arxiv | GBT | 70.66±0.41 | 70.52±0.36 | 71.49±0.32 |
|  | GraphCL | 70.63±0.32 | 71.42±0.41 | 71.19±0.21 |
|  | BGRL | 71.34±0.17 | 72.64±0.34 | 73.11±0.34 |

Table 2: The impact of different feature augmentation prompts on GBT, GraphCL, and BGRL.

We now examine the impact of different feature augmentation prompts (i.e., SAS, IDR, and SAR) on the ChatGAug variant. Table 2 presents the results for the PubMed and Arxiv datasets. Our observations are as follows: ⑨ **Reasoning-based prompts tend to outperform text summarization prompts.** Specifically, both IDR and SAR variants generally outperform SAS, particularly on Arxiv dataset. This is because IDR and SAR instruct ChatGPT to make predictions regarding potential label categories. This guidance encourages ChatGPT to establish connections between keywords in the textual attributes and abstract semantic meanings (i.e., potential categories). ⑩ **The best prompt templates could vary across different datasets.** Neither IDR nor SAR consistently outperforms the other, as demonstrated in Table 2. For example, BGRL achieves its best result on PubMed using the IDR prompt, while GBT achieves its best performance with the SAR prompt. In the future, we will investigate whether there potentially exists a universally suitable prompt template for GCL.

### 4.5 ADDITIONAL INVESTIGATION

In this section, we conduct further experiments to provide a better understanding of LLM4GCL. For comprehensive details on the experimental setups, please refer to Appendix D.

To answer **RQ4**, we explore the influence of different language model backbones on GCL methods. Figure 5 and Figure 12 in Appendix illustrate the outcomes obtained with BERT, DeBERTa, and RoBERTa backbones, revealing that **the optimal language model configuration varies for different GCL methods across datasets.** As an instance, BGRL attains the highest performance on PubMed when utilizing RoBERTa, whereas GraphCL achieves the best results with DeBERTa. Hence, defining a unified language model that exhibits strong generalization across various methods and datasets represents a promising avenue for future research.

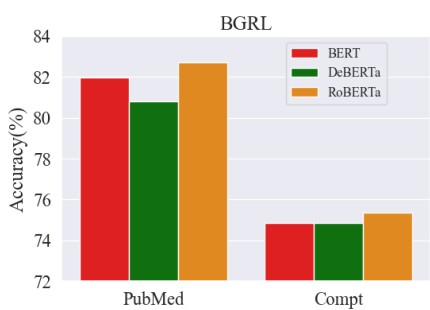

Figure 5: The impact of different PLM backbones on BGRL.

To answer **RQ5**, we select two representative backbones: GraphMAE (Hou et al., 2022) and S2GAE (Tan et al., 2023). In our experiments, we test their performance by replacing the original shallow embeddings with the ones generated by our fine-tuned PLMs, specifically focusing on DeBERTa. The results in Table 13 in Appendix showcase that **GraphMAE and S2GAE generally yield superior results when utilizing the node features generated by GIANT and TACL**. These findings align with our discoveries on GCL in Section 4.3, highlighting the potential applicability of LLM4GCL within the broader self-supervised graph learning domain.

**Embedding visualization.** For **RQ6**, we employ the t-SNE tool to visualize the extracted node features. The results, available in Figures 8& 9& 10 in the Appendix, illustrate the impact of different fine-tuning strategies. We observe from the visualization that the embedding space learned by GIANT and TACL excels in producing more distinct clusters. These outcomes provide additional support for the superior performance of GIANT and TACL in our primary experiments.

### 5 CONCLUSION

This study investigates contrastive learning to text-attributed graphs (TAGs) and introduces a comprehensive framework, LLM4GCL. LLM4GCL leverages large language models (LLMs) to enhance graph contrastive learning methods. Our contribution is twofold. Firstly, we present a novel approach, termed LLM-as-GraphAugmentor, which utilizes LLMs to augment graphs at both the feature and structural levels through the application of prompts. Secondly, we explore a suite of fine-tuning strategies designed to adapt pre-trained language models for encoding of textual attributes in TAGs, referred to as LLM-as-TextEncoder. Based on these two pipelines, we conduct extensive experiments across five diverse TAG datasets. We hope our empirical results can highlight the promising applications of LLMs in graph self-supervised learning and inspire future research.

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
