# A  DATASETS

To evaluate the proposed LLM4GCL framework, we conduct experiments on five benchmark datasets in Table 3. We list their detailed information below.

- **PubMed** (Sen et al., 2008) is a citation network commonly used in prior GSL works, with nodes representing papers and edges representing papers' citation relationships. Node features are bag-of-wards feature vectors by default. The label of each node is its category of research topic.

- **Ogbn-arxiv (Arxiv)** (Hu et al., 2021b) is a directed graph that represents the citation network between all computer science arXiv papers. Each node is an arXiv paper, and each directed edge indicates that one paper cites another one.

- **Electronics-Computers (Compt)** and **Electronics-Photo (Photo)** (McAuley et al., 2015) are segments of the Amazon co-purchase graph, where nodes represent goods, edges indicate that two goods are frequently bought together, node features are bag-of-words encoded product reviews by default, and class labels are given by the product category.

- **Books-History (Hist)**. Each node represents a book, edges indicate that two books are frequently bought together and class labels are given by the book category.

| Data | # Nodes | # Edges | # Features | # Classes | # AverageText Length |
|---|---|---|---|---|---|
| PubMed | $19,717$ | $44,338$ | 500 | 3 | 1649.25 |
| Ogbn-Arxiv | 169343 | $1,166,243$ | 128 | 40 | 1177.993 |
| Books-History | $41,551$ | $400,125$ | 768 | 12 | 1427.397 |
| Electronics-Computers | $87,229$ | $808,310$ | 768 | 10 | 492.767 |
| Electronics-Photo | $48,362$ | $549,290$ | 768 | 12 | 797.822 |

Table 3: Dataset statistics of five text-attributed graphs (TAGs).

We focus on semi-supervised classification and split PubMed and Arxiv datasets according to their standard data split provided in (Kipf & Welling, 2016; Hu et al., 2020a), i.e., 20 shot samples for each class on PubMed (Kipf & Welling, 2016) and the official data split from (Hu et al., 2020a): 54%/18%/28% for the train/validation/test sets. For Compt, Photo, and Hist datasets, we follow the few-shot learning setting (Garcia & Bruna, 2017) to randomly generate 300/30/1000 labeled nodes per class for the train/validation/test sets. All results are the average of ten independent runs.

# B  EXPERIMENTAL CONFIGURATION

In this section, we will detail the hyperparameters adopted for each method used in our experiments.

## B.1  BASELINE METHODS

We introduce several baseline methods. In the subsequent sections, each method will be described in detail.

### B.1.1  GRAPH NEURAL NETWORKS

- **GCN** (Kipf & Welling, 2016). Two GCN layers are applied. Batch normalization (Ioffe & Szegedy, 2015) is applied on the first hidden layer. The activation is ReLU (Nair & Hinton, 2010).

- **GAT** (Veličković et al., 2018a) extends the Graph Convolutional Network (GCN) by leveraging masked self-attentional layers to handle graph-structured data, allowing nodes to attend over their neighborhoods' features with varying weights.

Table 4 summarizes the hyperparameters used for GCN and GAT.

| Hyperparameters | GCN | GAT |
|---|---|---|
| #GNN layers | 2 | 2 |
| hidden dim | 64 | 64 |
| learning rate | 0.01 | 0.01 |
| weight decay | 0.0005 | 0.0005 |

Table 4: Hyperparameters for the GCN and GAT models.

### B.1.2 GRAPH CONTRASTIVE LEARNING METHODS

- **GraphCL** (You et al., 2020) is a method that leverages data augmentation to facilitate self-supervised learning on graphs. Through various augmentation strategies, it generates different views of the original graph, and then learns node representations by minimizing the contrastive loss between the augmented views. In our experiment, the augmentation strategy is set to random mask, the batch size is set to 1, the number of epochs is set to 10000, the learning rate is set to 0.001, and the hidden dim is set to 156.

- **BGRL** (Thakoor et al., 2021) is a method that learns graph representations by predicting alternative augmentations of the input graph. It employs a two-encoder setup where one encoder processes the original graph and the other processes an augmented version of the graph. The learning objective is to minimize the divergence between the representations produced by the two encoders. Table 5 provides an overview of the hyperparameters of different datasets for BGRL.

- **GBT** (Bielak et al., 2022) is a self-supervised graph representation learning method, which utilizes a cross-correlation-based loss function instead of negative sampling. Table 6 provides an overview of the hyperparameters of different datasets for GBT.

| Hyperparameters | PubMed | Ogbn-Arxiv | Hist | Compt | Photo |
|---|---|---|---|---|---|
| epochs | 1000 | 1000 | 500 | 500 | 500 |
| warmup epochs | 10 | 200 | 1 | 1 | 1 |
| edge dropout 1 | 0.4 | 0.6 | 0.3 | 0.3 | 0.3 |
| node feature dropout 1 | 0 | 0.3 | 0.3 | 0.3 | 0.3 |
| edge dropout 2 | 0.5 | 0.6 | 0.2 | 0.2 | 0.2 |
| node feature dropout 2 | 0 | 0.4 | 0.4 | 0.4 | 0.4 |
| learning rate | 5e-4 | 1e-2 | 1e-2 | 1e-2 | 1e-2 |
| weight decay | 1e-5 | 1e-5 | 1e-5 | 1e-5 | 1e-5 |

Table 5: Hyperparameters for BGRL on different datasets.

| Hyperparameters | PubMed | Ogbn-Arxiv | Hist | Compt | Photo |
|---|---|---|---|---|---|
| total epochs | 1000 | 700 | 4000 | 4000 | 4000 |
| warmup epochs | 200 | 100 | 400 | 400 | 400 |
| log interval | 1000 | 100 | 1000 | 1000 | 1000 |
| embedding dimension | 256 | 256 | 256 | 256 | 256 |
| learning rate | 1.e-3 | 1.e-3 | 5.e-4 | 1.e-3 | 5.e-4 |
| mask feature rate | 0.3 | 0 | 0.1 | 0.1 | 0.1 |
| mask edge rate | 0.4 | 0.4 | 0.5 | 0.5 | 0.5 |

Table 6: Hyperparameters for GBT on different datasets.

### B.1.3 GRAPH GENERATIVE LEARNING METHODS

- **GraphMAE** (Hou et al., 2022) is a masked graph autoencoder that focuses on feature reconstruction with both a masking strategy and scaled cosine error that benefit the robust training process. In our experiment, the mask rate is set to 0.5, the input feature dropout

is set to 0.2, the attention dropout is set to 0.2, the number of hidden layers is set to 2, the number of hidden units is set to 512, the learning rate is set to 0.001, learning rate for evaluation is set to 0.01, the number of attention heads is set to 4.

- **S2GAE** (Tan et al., 2023) unleashes the power of GAEs with minimal nontrivial efforts. It randomly masks a portion of edges and learns to reconstruct these missing edges with an effective masking strategy and an expressive decoder network. In our experiment, the number of hidden layers is set to 2, the number of decode layers is set to 3, the dropout is set to 0.5, the batch size is set to 1024, and the learning rate is set to 0.001.

### B.1.4 ARCHITECTURE OF LARGE LANGUAGE MODELS

- **BERT** (Devlin et al., 2018) is designed to pre-train deep bidirectional representations from unlabeled texts by jointly conditioning on both left and right context in all layers.

- **RoBERTa** (Liu et al., 2019) is a robustly optimized BERT pre-training approach, which uses a larger text dataset for pre-training and employs a dynamic mask strategy for pre-training objectives.

- **DeBERTa** (He et al., 2020) improves the BERT and RoBERTa models using a disentangled attention mechanism and an enhanced mask decoder.

Table 7 provides an overview of the hyperparameters used for BERT, RoBERTa, and DeBERTa. Other hyperparameters are set to the default values.

For the sake of experiment reproducibility, we have employed Graph Neural Network (GNN) implementations from the PyG package by (Fey & Lenssen, 2019). To implement the Graph Contrastive Learning (GCL) methods, namely GraphCL, BGRL, and GBT, we meticulously followed the procedures elucidated in (Zhu et al., 2021a). For feature extraction, we harnessed Pre-trained Language Models (PLMs) from Huggingface and employed OpenAI's ChatGPT 3.5 for graph augmentation. Comprehensive configurations for Graph Neural Networks (GNNs), GCL, and language models can be found in Table 4, Table 5& 6, and Table 7, respectively.

When dealing with different LLM4GCL variants, we observed that various datasets and GCL backbones often exhibit distinct preferences regarding model-level hyperparameters and language model options. Consequently, for each evaluation scenario, we conducted a thorough search for the optimal experimental configuration within the suggested parameter ranges outlined in the original papers, taking into account the specific language model in use, such as BERT, DeBERTa, and RoBERTa.

| Hyperparameters | BERT | RoBERTa | DeBERTa |
|---|---|---|---|
| learning rate | 2e-5 | 2e-5 | 2e-5 |
| dropout | 0.3 | 0.3 | 0.3 |
| attention dropout | 0.1 | 0.1 | 0.1 |
| weight decay | 0 | 0 | 0 |

Table 7: Hyperparameters for the BERT, RoBERTa and DeBERTa models.

### B.2 RESULT ANALYSIS

### B.2.1 WHY GRAPH CONTRASTIVE LEARNING ON TAGS?

Graph Contrastive Learning (GCL) on Text-Attributed Graphs (TAGs) emerges as a practical approach due to the rich and complex data represented in TAGs, which encapsulate a wealth of semantic information through textual attributes associated with each node. This approach aligns with the self-supervised learning paradigm, which is particularly indispensable in real-world settings where label acquisition is particularly expensive or impractical. The LLM4GCL framework, as discussed, introduces a way to make use of Large Language Models (LLMs) for both feature and structural augmentation of the graph as demonstrated, yielded richer node representation.

We conducted few-shot learning experiments to further assess the effectiveness of the proposed fine-tuning strategies in graph augmentations. The results are illustrated in Figure 6 and Figure

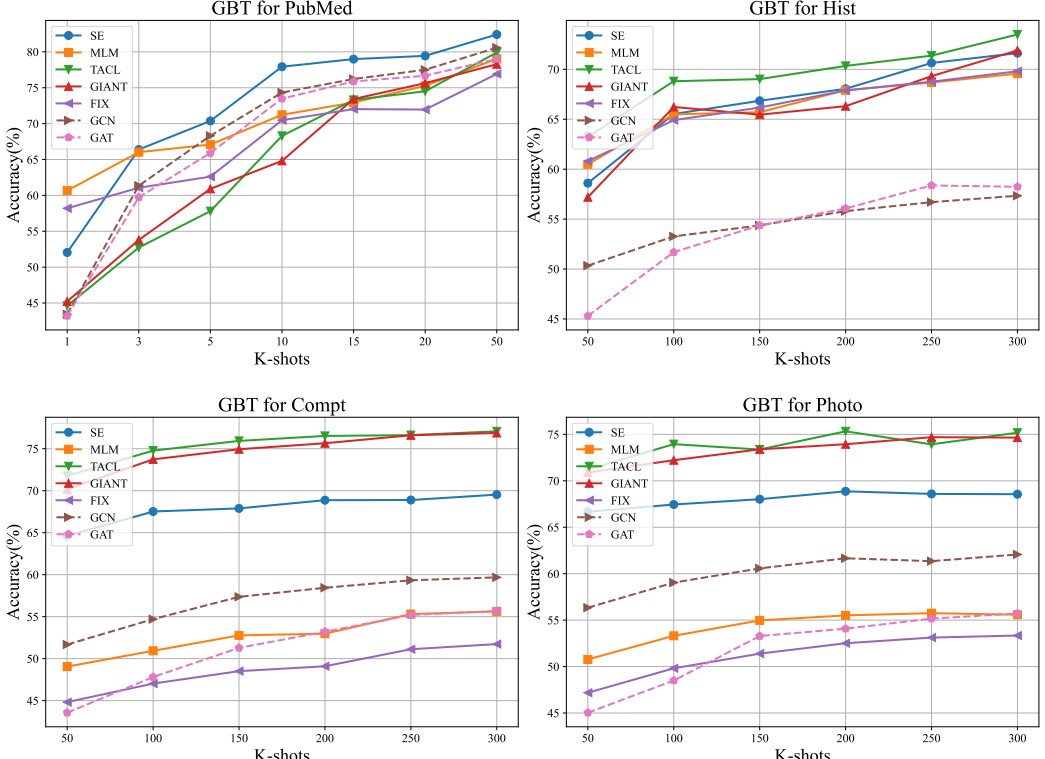

Figure 6: Few-shot learning results on GBT.

7. We established our baseline using two Graph Contrastive Learning methods, namely GraphCL and GBT. For feature augmentation, we employed various strategies including SE, MLM, TACL, GIANT, FIX, GCN, and GAT across diverse datasets namely Pubmed, Photo, Compt, and Hist. Our experiments, depicted in the figures, indicates the employing LLMs can lead to an improvement in the model's accuracy, thus showcasing their potential in enhancing the graph learning process.

### B.2.2 CAN LLMS ENHANCE THE ENCODING OF TEXTUAL ATTRIBUTES?

LLMs capture semantic meaning and context in a much more sophisticated way than many other traditional embedding methods, especially on Text-Attribute Graphs (TAGs), which have abundant textual information and intricate node relationships. Also, LLMs can be fine-tuned with the datasets we mention in this paper, allowing them to adapt to the characteristics of the TAGs. To verify the effectiveness of LLMs to enhance the encoding of textual attributes, we employ three different unsupervised fine-tuning strategies. In this appendix, we present more details about these experiments. This experiment is conducted on Nvidia A100 GPU with 80GB memory. Table 8 provides

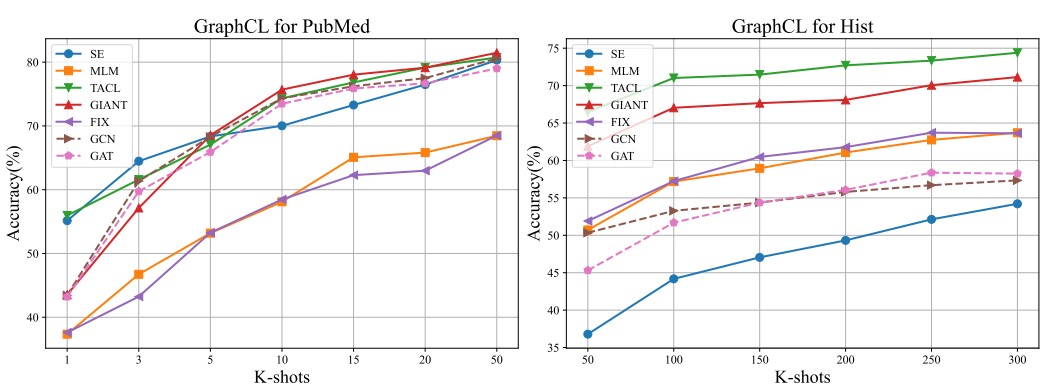

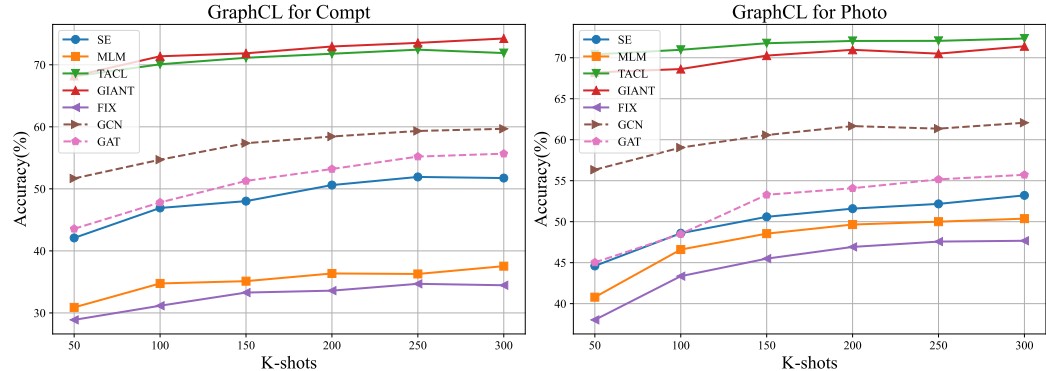

Figure 7: Few-shot learning results on GraphCL.

the overview of hyperparemeters for MLM and TACL. The hyperparameters of GIANT are set to the default values of XTransformer (Liu & Chen, 2022).

| Hyperparameters | MLM | TACL |
|---|---|---|
| learning rate | 5e-4 | 2e-5 |
| dropout | 0.3 | 0.3 |
| attention dropout | 0.1 | 0.1 |
| weight decay | 0.01 | 0 |
| batch size | 9 | 32 |
| epochs | 4 | 2 |

Table 8: Hyperparameters for the MLM and TACL.

After fine-tuning these three models, we test the semi-shot learning accuracy on BGRL, GraphCL and GBT with the text embedding generated by the fine-tuned models. For **PubMed**, we set the K list as [1,3,5,10,15,20,50]. For **Arxiv** and **Amazon Datasets**, we set the K list as [50, 100, 150, 200, 250, 300].

### B.2.3 HOW EFFECTIVE IS LLMS IN PERFORMING GRAPH AUGMENTATIONS?

In this appendix, we delve deeper into the specifics of these experiments. To begin, we employ feature augmentation prompts, such as SAS, IDR, and SAR (depicted in Figure 2 (left)), to generate augmented textual attributes for each node. Subsequently, we employ the most suitable pre-trained language model, as identified in the LLM-as-TextEncoder section, tailored to the distinct characteristics of different datasets, to serve as the text encoder. Utilizing the PLM backbone and the augmented textual attributes, we create a new feature matrix denoted as $\hat{\mathbf{X}}$. Simultaneously, we apply the graph structure prompt, as discussed in Figure 2 (right), to generate augmented graph structure $\hat{\mathbf{A}}$, leading to the formation of an augmented graph $\hat{\mathcal{G}} = (\hat{\mathbf{X}}, \hat{\mathbf{A}})$.

Following this augmentation process, we can seamlessly integrate $\hat{\mathcal{G}}$ into the standard GCL pipeline, replacing one of the augmented views. Typically, existing GCL methods perform the following steps to generate two augmented graphs in each iteration when given an input graph $\mathcal{G}$:

$$\begin{aligned} \mathcal{G}_1 &= (\tau_f(\mathbf{X}), \tau_s(\mathbf{A})) \\ \mathcal{G}_2 &= (\tau_f(\mathbf{X}), \tau_s(\mathbf{A})). \end{aligned} \tag{4}$$

However, with our augmented graph $\hat{\mathcal{G}}$ generated using LLMs, we can train the GNN encoder to produce similar representations between the augmented view $\mathcal{G}_1$ and $\hat{\mathcal{G}}$. The distinction lies in the fact that we only need to perform stochastic feature and structure augmentations once during each iteration, while maintaining the second augmented view as a fixed entity represented by $\hat{\mathcal{G}}$.

# C   LLM FOR GRAPH CONTRASTIVE LEARNING

## C.1   LLM FOR NODE AUGMENTATION

The conventional augmentation of a node's feature vector in Graph Contrastive Learning (GCL) through stochastic functions has its limitations, as it can not effectively manipulate the inherent textual attribute $S_v$. To address this, we propose using Large Language Models (LLMs) to directly perturb the input text $S_v$ to derive an augmented textual attribute $\hat{S}_v$. Three distinctive prompt templates are introduced: Structure-Aware Summarization (SAS), Independent Reasoning (IDR), and Structure-Aware Reasoning (SAR), each with a unique approach towards leveraging semantic and structural information for augmentation. To mitigate the query overhead associated with Chat-GPT, we adopt a strategy of randomly sampling 10 neighbors for each anchor node when crafting structure-aware prompts (SAS and SAR) in the experimental setup. This methodology aims to enrich the augmentation of textual attributes, potentially enhancing the GCL framework's capability in capturing semantic relationships within the graph structure.

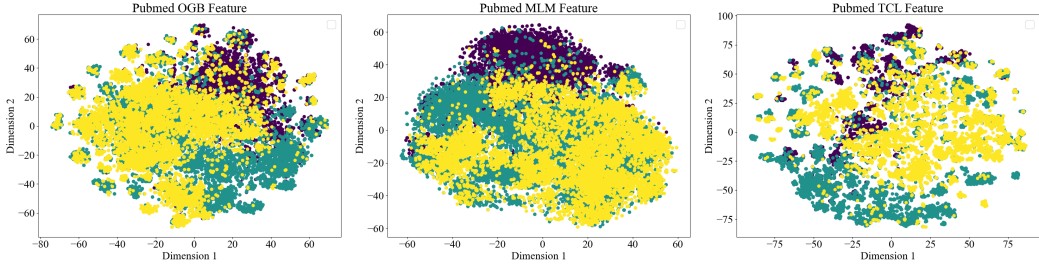

Figure 8: The 2-dimensional embedding visualization with t-SNE on PubMed dataset.

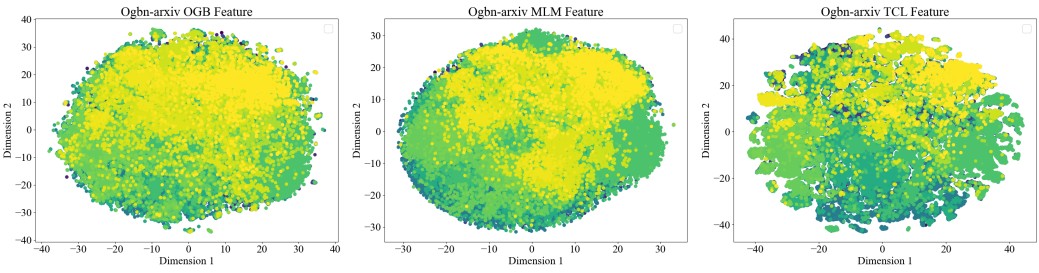

Figure 9: The 2-dimensional embedding visualization with t-SNE on Arixv dataset.

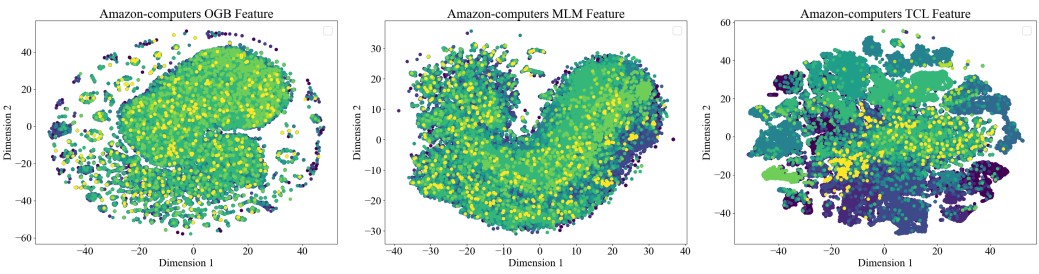

Figure 10: The 2-dimensional embedding visualization with t-SNE on Compt dataset.

---

**Algorithm 1** Graph Structure Augmentation

---

1: **procedure** STRUCTURE_AUGMENTATION($G, v, A, LLM$)
2:     $N_v \leftarrow$ ConnectedNodes(v)
3:     $\bar{N}_v \leftarrow$ DisconnectedNodes(v)
4:
5:     *// Employ shallow embedding algorithms for initial similarity assessment*
6:     $N_v \leftarrow$ SelectTopK($N_v$, 20, descending)
7:     $\bar{N}_v \leftarrow$ SelectTopK($\bar{N}_v$, 20, ascending)
8:     prompt_connect $\leftarrow$ CreatePrompt($v, N_v$)
9:     prompt_disconnect $\leftarrow$ CreatePrompt($v, \bar{N}_v$)
10:    candidates_connect $\leftarrow$ LLM(prompt_connect)
11:    candidates_disconnect $\leftarrow$ LLM(prompt_disconnect)
12:
13:    *// Update adjacency matrix based on LLM decisions*
14:    **for** each node $u$ in v **do**
15:        Update $\hat{A}[v][u]$ based on candidates_connect and candidates_disconnect
16:    **end for**
17:
18:    **return** $\hat{A}$
19: **end procedure**

---

## C.2   LLM FOR GRAPH AUGMENTATION

In this appendix, we present the pseudocode for LLM as Graph Augmentation. The algorithm `Structure_Augmentation` takes as input a graph $G$, a node $v$, an adjacency matrix $A$, and an LLM. It returns an augmented adjacency matrix $\hat{A}$ reflecting the updated graph structure.

Initially, the sets $N_v$ and $\bar{N}_v$ are constructed to represent the connected and disconnected nodes to node $v$, respectively. To enhance query efficiency, shallow embedding algorithms such as Bag of Words (BoW) are employed initially to assess the pairwise similarity between nodes. Following this, the top 20 nearest neighbors from $N_v$ and the top 20 unconnected nodes from $\bar{N}_v$ are selected based on their similarity scores, in descending and ascending order respectively, for querying the LLM in subsequent steps. Following this, two prompts are created, one for connected Nodes $N_v$ (prompt_connect), and another for disconnected nodes (prompt_disconnect). This is crucial as it helps in addressing the connections and disconnections separately. Subsequently, the LLM is queried with these prompts to obtain decisions on whether each node in $N_v$ (or $\bar{N}_v$) should be connected (or disconnected) to $v$. This step leverages the language model's ability to interpret and generate informed decisions based on textual attributes. Based on the responses from the LLM, the adjacency matrix $\hat{A}$ is updated. If the decision is to connect a previously disconnected node $u$ to node $v$, the corresponding entry in $\hat{A}$ is set to 1.

This approach ensures that the graph augmentation process is systematically carried out, leveraging the LLM's capability to understand and manipulate textual attributes for enhancing the graph's structure and features.

## C.3   LLM FOR FEATURE EXTRACTION

We will demonstrate the use of LLMs for feature extraction. Three fine-tuning strategies are introduced.

- **Masked Language Modeling (MLM):** In MLM (See Figure 11), a portion of the tokens in the textual attributes is masked (hidden), and the model is trained to predict these masked tokens based on the context provided by the remaining unmasked tokens. This strategy helps in fine-tuning the model on textual attributes, thus improving its understanding and representation of the textual data.

- **Topology-Aware Contrastive Learning (TACL):** TACL (See Figure 11) is aimed at understanding the topology of the graph by employing contrastive learning. In this approach,

the model learns to bring the representations of connected nodes closer in the embedding space while pushing the representations of unconnected nodes further apart.

- **Multi-Scale Neighborhood Prediction (GIANT):** GIANT (See Figure 11) adopts an extreme multi-label classification technique to reconstruct all neighbors of a node. By employing a hierarchical node cluster tree, this strategy addresses the computational challenges posed by the high-dimensional output space. It aims at transforming the reconstruction task into a multi-label classification problem, thus providing a computationally feasible solution while capturing the neighborhood structure of nodes.

By using LLMs, the learning in GCL methods is enhanced by addressing the limitations of traditional approaches, especially in effectively utilizing textual attributes for both feature and structure augmentations. By employing these three fine-tuning strategies, we can efficiently adapt a PLM model to TAGs.

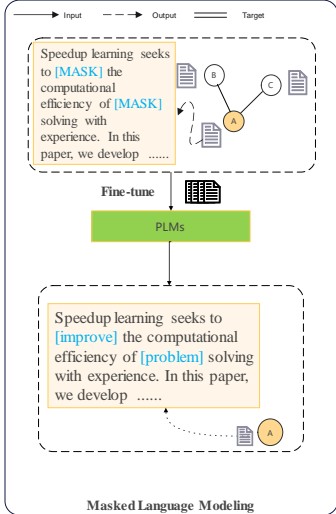 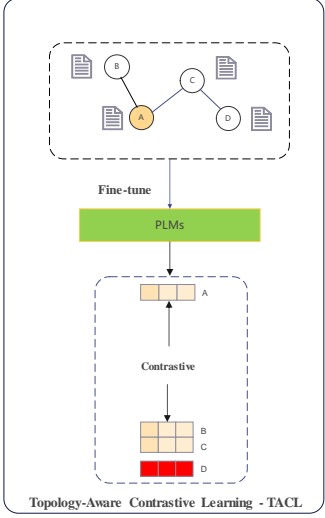 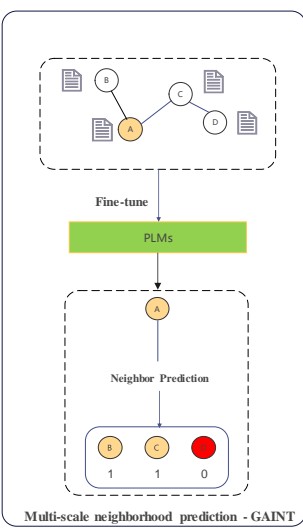

Figure 11: LLM-as-TextEncoder. **Left**: Standard masked language modeling (MLM). **Middle**: Topology-aware contrastive learning (TACL). **Right**: Multi-scale neighborhood prediction (GIANT).

# D    ADDITIONAL EXPERIMENTS

In this appendix, we conduct additional experiments to further investigate the proposed method.

**Which types of language models are best suited for GCL on TAGs?** We investigated the impact of various language model backbones on GCL methods. In our experiments, we adhered to the best-performing experimental configurations for BGRL, GraphCL, and GBT, as outlined in Section 4.3. We evaluated their performance using different language model backbones, including BERT, DeBERTa, and RoBERTa. For instance, when employing DeBERTa, our approach involved two main steps. Initially, we fine-tuned DeBERTa using the TACL strategy on the textual attributes of Text-Attributed Graphs (TAGs). Subsequently, we utilized the resulting DeBERTa model to encode the original textual attributes, extracting features from them. Following this feature extraction step, all GCL methods leveraged these generated node features as input for their respective training processes.

Figure 5 and Figure 12 in Appendix illustrates the outcomes obtained with BERT, DeBERTa, and RoBERTa backbones, revealing that **the optimal language model configuration varies for different GCL methods across diverse evaluation datasets.** As an instance, BGRL attains the highest performance on PubMed when utilizing RoBERTa, whereas GraphCL achieves the best results with DeBERTa. Hence, defining a unified language model that exhibits strong generalization across various methods and datasets represents a promising avenue for future research.

Table 9: Prompt templates of independent reasoning for various datasets.

| Dataset | Prompt Template |
|---|---|
| Cora
PubMed
Arxiv | Input:
Central Node: Title [title text] Abstract [abstract text]

Instruction:
Which of the following subcategories of [field] does this paper belong to:
[categories]? If multiple options apply, provide a comma-separated list
ordered from most to least related. For each choice you gave,
explain how it is present in the text. |
| Amazon-History | Input:
Main book: [Description and title]

Instruction:
Using the provided history-related book's title and description,
categorize the main book into one of the following categories:
['World','Americas','Asia','Military','Europe','Russia','Africa',
'Ancient Civilizations', 'Middle East','Historical Study and
Educational Resources','Australia and Oceania','Arctic and Antarctica'].
Please provide your reasoning.

Respond in this format:
The book belongs to the [Category] category due to
[evidence from the book product descriptions]. |
| Amazon-Computers
Amazon-Photo | Input:
Main product: [Review]

Instruction:
Given the product review provided, please categorize
the product into one of the following categories: [categories]
Your classification should be based on the content of the review.
Please support your answer with evidence from the review.

Response Format:
The product falls under the category
of [Category]. This determination is based on the product review,
where [specific details from the review supporting the classification]. |

**Can the proposed LLM4GCL enhance the performance of generative-based self-supervised learning methods?** To answer this question, we select two representative backbones for experimentation: Graph-MAE (Hou et al., 2022) and S2GAE (Tan et al., 2023). Both of them operate by randomly masking portions of the input graph and subsequently learning to reconstruct these masked components. In our experiments, we evaluate their performance by substituting the original shallow embeddings with the embeddings generated by our fine-tuned PLMs.

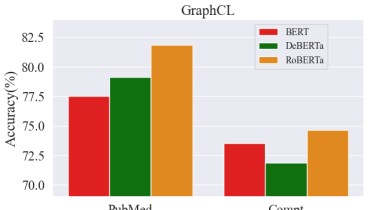

Figure 12: The impact of different PLM backbones on GraphCL.

Specifically, we test their performance by replacing the original shallow embeddings with the advanced one generated by our fine-tuned PLMs, specifically focusing on DeBERTa. We search the best model configurations of GraphMAE and S2GAE according to their papers. The results in Table 13 showcase that **GraphMAE and S2GAE generally yield superior results when utilizing the node features generated by GIANT and TACL**. These findings align with our discoveries in Section 4.3, highlighting the potential applicability of LLM4GCL within the broader self-

Table 10: Prompt templates of struct-aware summarization for various datasets.

| Dataset | Prompt Template |
|---|---|
| Cora
PubMed
Arxiv | Input:
Central Node: Title [title text] Abstract [abstract text]
Linked Nodes: Title$_i$ [title text] Abstract$_i$ [abstract text]

Instruction:
Please use the information from the central node text attribute
and the linked papers' text attribute to revise the text of the
central paper. Feel free to rephrase, restructure, or expand upon
the existing abstract as needed, yet reply in one paragraph. |
| Amazon-History | Input:
Main product: [Description]
Product$_i$ frequently purchased together:[Description of product$_i$]

Instruction:
Using the provided product description and, if available,
descriptions of associated products, provide a concise summary of
the main product's key features and benefits.

Respond Format: "The main product can be summarized as:
[Concise Summary] due to [evidence from descriptions]." |
| Amazon-Computers
Amazon-Photo | Input:
Main product: [Review]
Products frequently purchased together:[Review of product$_i$]

Instruction:
Based on the product review provided, please summarize the
main points or sentiments expressed by the reviewer.
Your summary should capture the essence of the review in a concise manner.
Please support your answer with evidence from the review.

Response Format:
The review highlights that [Summary].
This summary is derived from the reviewer's comments, where
[specific details from the review supporting the summary] are mentioned. |

supervised graph learning domain, including contrastive (Zhu et al., 2021a) and generative (Zhang et al., 2022; Li et al., 2022) approaches.

**Embedding visualization.** We employed the t-SNE tool to visualize the extracted node features resulting from different fine-tuning strategies. Figure 8, Figure 9, and Figure 10 showcase the visualization results on the PubMed, Arxiv, and Photo datasets, respectively. These figures reveal a noteworthy observation: the structure-aware fine-tuning strategy, exemplified by TACL, excels in producing more distinct clusters within the latent feature space, which is known to be effective for follow-up classification tasks (Yang et al., 2016; Asano et al., 2019; Zhu et al., 2020b). These outcomes provide additional support for the superior performance of GIANT and TACL in our primary experiments.

Table 11: Prompt templates of structure-aware reasoning for various datasets.

| Dataset | Prompt Template |
|---|---|
| Cora
PubMed
Arxiv | Input:
Central Node: Title [title text] Abstract [abstract text]
Candidate Papers: Title$_i$ [title text] Abstract$_i$ [abstract text]

Instruction:
Please use the information from the central node
text attribute and the linked papers' text attribute
to revise the text of the central paper. Feel free
to rephrase, restructure, or expand upon the existing
abstract as needed, yet reply in one paragraph. |
| Amazon-History | Input:
Main book: [Description and Title]
Books about History:[Description$_i$ and Title$_i$]

Instruction:
Using the provided product description and,
if available, descriptions of associated products,
categorize the main product into one of the categories: [categories]

Respond Format:
"The product belongs to the [Category] category due to
[evidence from descriptions]." |
| Amazon-Computers
Amazon-Photo | Input:
Primary product review: [Review]
List of other product reviews: [Reviews]

Instruction:
Given the product review provided, please categorize the product
into one of the following categories: [categories]
Your classification should be based on the content of the review.
Please support your answer with evidence from the review.

Response Format:
The product falls under the category of [Category].
This determination is based on the product review,
where [specific details from the review supporting the classification]. |

Table 12: Prompt templates of graph structure augment for various datasets.

| Dataset | Prompt Template |
|---------|-----------------|
| Cora
PubMed
Arxiv | Input:
Anchor Paper: Title [title text] Abstract [abstract text]
Candidate Papers: Title$_i$ [title text] Abstract$_i$ [abstract text]

Instruction:
Can you help me to evaluate whether the anchor paper and each
candidate paper should be connected or cited together based on
the content of their respective abstracts? Let's think step by step.
First, we need to know how many candidate abstracts are there.
From the candidate abstracts list, we know that
there are [length] candidate abstracts.
Second, we need to decide for each candidate abstract whether it can be
connected to the anchor paper or not. 1 means yes and 0 means no.

Reply in this format: Candidate List []. |
| Amazon-History | Input:
Main book: [Description and Title]
Books about History:[Description$_i$ and Title$_i$]

Instruction:
Analyze the main book title and description alongside the list
of books title and descriptions. Identify which books share thematic
similarities or cater to the same historical interests as the main book and
are likely to be frequently purchased together by readers with an interest
in similar historical topics or eras.
Return a binary list where '1' denotes a likely paired purchase
with the main book,and '0' suggests otherwise and the length of the list
should match the length of the "List of book titles and descriptions,
which is [length].

Expected Output:
[1,0,1,....,0] (A list of 1s and 0s corresponding
to each book in the "List of product reviews") |
| Amazon-Computers
Amazon-Photo | Input:
Primary product review: [Review]
List of other product reviews: [Reviews]

Instruction:
Based on the content of the primary product review and the list of products
reviews, determine if the primary product and the products in the list are
likely to be frequently purchased together.
Return a list of 1s and 0s, where the length of the list should match
the length of the List of product reviews, which is [length].
1 indicates that the respective product from the list is likely to be
purchased with the primary product, and 0 indicates otherwise.

Expected Output:
[1,0,1,....,0] (A list of 1s and 0s corresponding
to each product in the "List of product reviews") |

| Feature type | GIANT | | MLM | | TACL | |
|---|---|---|---|---|---|---|
| method | GraphMAE | S2GAE | GraphMAE | S2GAE | GraphMAE | S2GAE |
| PubMed | 77.74±2.81 | 78.43±3.61 | 47.08±8.21 | 42.46±7.30 | 79.45±0.97 | 79.39±0.26 |
| Arixv | 72.58±0.00 | 70.91±0.01 | 21.56±0.00 | 54.29±0.02 | 70.33±0.02 | 71.04±0.14 |
| Amazon-computers | 73.91±0.17 | 84.85±0.22 | 9.08±6.56 | 69.56±0.46 | 75.45±0.16 | 82.20±0.29 |
| Amazon-history | 75.59±0.62 | 73.56±0.92 | 11.43±12.25 | 69.03±0.28 | 75.65±0.67 | 69.06±1.01 |
| Amazon-photo | 71.66±0.48 | 77.89±0.48 | 6.91±3.89 | 69.69±0.10 | 71.97±0.38 | 76.42±0.46 |

Table 13: Semi-supervised accuracy results for generative-based self-supervised learning methods.