# OpenReview forum: "LLM4GCL: CAN LARGE LANGUAGE MODEL EM-POWER GRAPH CONTRASTIVE LEARNING?"
_ICLR.cc/2024/Conference — Submitted to ICLR 2024_

### Official Review · Reviewer_nivK · 2023-10-12

**Soundness:** 2 fair
**Presentation:** 2 fair
**Contribution:** 2 fair
**Rating:** 3
**Confidence:** 4

**Summary:**

This paper focuses on the research topic: Large Language Models for Graph Contrastive Learning, which addresses the limitations of the existing method. The method is conducted on Text-Attributed Graphs (TAGs) and proposes two comprehensive approaches to enhance graph learning, i.e., LLM-as-GraphAugmentor, and LLM-as-TextEncoder.  The comprehensive experimental results leveraging LLMs in GCL for TAGs can lead to significant improvements in node classification tasks.

**Strengths:**

1. The study addresses a key limitation in existing graph learning methods by focusing on Text-Attributed Graphs (TAGs), acknowledging the challenges of converting textual data into numerical features.

2. The research explores two distinct pipelines - one using LLMs to augment graphs directly and another to encode textual attributes into embeddings. This comprehensive approach provides a thorough investigation of potential solutions.

3. The findings indicate that LLM4GCL holds promise in improving the performance of state-of-the-art graph learning methods, highlighting the practical significance of this research.

**Weaknesses:**

1. On one hand, graph contrastive learning can adapt more unlabeled data for self-supervised learning. On the other hand, the API of LLM is quite expensive. It is a very high cost to do some analysis. I think the author should report how much money is required for each dataset. Moreover, the API of chatgpt has a limitation on the amount of queries per day. How long for one experiment on one API should also be discussed

2. There is no comparison between gpt for argumentation and directly using GPT to do the prediction. [1] show very good node classification results directly utilizing LLM for prediction

3. The Giant baseline is not well-discussed. There should be a clear discussion with GIANT, showing the difference and the contribution of the paper

4. The proposed method seems to be just comparable with GIANT while being much more expensive. Moreover, in table 1, GIANT on Hist can reach a performance of 74.16 which should be the second best method.

5. The selected datasets do not include the large-scale ogbn-product dataset.

6. The motivation for the TACL is still lacking. Why does this method fit in the LM than other GCL methods?

7. What is the novelty of the LLM as a text encoder? I think there already existing many works in this domain, e.g., [2, 3]

8. Figure 4 should be rescaled.

9. Lack of ablation study in Figure 4, there should be chatgpt Aug for GCL and directly training.


[1] He, Xiaoxin, et al. "Explanations as Features: LLM-Based Features for Text-Attributed Graphs." arXiv preprint arXiv:2305.19523 (2023).
[2] Chen, Zhikai, et al. "Exploring the potential of large language models (llms) in learning on graphs." arXiv preprint arXiv:2307.03393 (2023).
[3] Zhao, Jianan, et al. "Learning on Large-scale Text-attributed Graphs via Variational Inference." The Eleventh International Conference on Learning Representations. 2022.

**Questions:**

See the above weakness

---

### Official Review · Reviewer_aMBc · 2023-10-16

**Soundness:** 3 good
**Presentation:** 3 good
**Contribution:** 2 fair
**Rating:** 3
**Confidence:** 4

**Summary:**

This paper investigates the integration of Large Language Models (LLM) into the field of graph contrastive learning (GCL). Specifically, the authors explore the utilization of LLMs as Graph Augmentors, where they employ LLMs to modify node features through textual attribute revisions, independent reasoning, and structure-aware reasoning. Additionally, LLMs are used to manipulate the graph topology via structure-aware reasoning. The paper provides a thorough examination of the benefits of incorporating LLMs into the GCL frameworks for TAGs.

**Strengths:**

- The paper explores a relatively new area by applying Large Language Models (LLMs) to Graph Contrastive Learning (GCL). This approach is unique and addresses a growing research interest.

- The paper procides comprehensive experiments. It covers a wide range of GCL settings and node embedding methods, ensuring the reliability of the results.

- The proposed method of LLMs-as-GraphAugmentor and LLM-as-TextEncoder can be intergrated as a plug-in component for existing GCL methods.

**Weaknesses:**

- The performance of the proposed method falls short in comparison to relevant benchmarks. For instance, in the case of the arXiv dataset, the zero-shot accuracy of 73.50 achieved with GPT-3.5, as reported in [1], sets a baseline for comparison. The independent reasoning approach employed for node feature augmentation in this paper essentially replicates the zero-shot task, where the model is provided with the paper's title, abstract, and an instruction to classify the paper. Given this, the expected outcome should be at least matching the accuracy achieved by this straightforward baseline. However, the results on the arXiv dataset do not convincingly demonstrate the effectiveness of the proposed LLM-augmentation+ GCL method, especially if it doesn't outperform vanilla LLM performance. A potential solution could be to report the accuracy of node classification following both independent reasoning and structure-aware reasoning in the node feature augmentation.
- Similar concerns arise with the performance on the PubMed dataset, where the reported zero-shot accuracy is 93.42.

References:

[1] Harnessing Explanations: LLM-to-LM Interpreter for Enhanced Text-Attributed Graph Representation Learning, https://arxiv.org/abs/2305.19523

**Questions:**

Q1: The concept of "LLM-as-TextEncoder" is somewhat misleading. It appears that PLMs are fine-tuned and then used to embed text features into continuous ones, instead of using LLMs.


Q2: In Table 2, when analyzing the impact of different feature augmentation prompts, could you clarify whether graph structure augmentation is activated? It seems that for the arXiv dataset, there is no significant improvement. For instance, the best accuracy is 73.11±0.34 (BGRL+SAR), which doesn't appear to outperform the baseline BGRL+SAR without LLM augmentation (73.28±0.15). Could you offer insights into why this outcome was observed?

---

### Official Review · Reviewer_GqWv · 2023-11-01

**Soundness:** 2 fair
**Presentation:** 2 fair
**Contribution:** 2 fair
**Rating:** 3
**Confidence:** 4

**Summary:**

This paper explores the potential of large language models (LLMs) in enhancing graph contrastive learning (GCL) on text-attributed graphs (TAGs). The authors propose two approaches: LLM-as-GraphAugmentor and LLM-as-TextEncoder, which leverage LLMs to augment graphs at both the feature and structural levels through the application of prompts. Through comprehensive and systematic studies on various benchmark datasets, the authors demonstrate the good results of their approach.

**Strengths:**

1. This paper explores LLM on graph constrastive learning, which is promising.
2. The paper explores fine-tuning strategies designed to adapt pre-trained language models for encoding of textual attributes in TAGs.
3. The paper presents various studies on six benchmark datasets

**Weaknesses:**

1. The novelty and application are limited
2. The experiments are weak

**Questions:**

1. The novelty and application are limited.  This paper proposes fine-tuning and masking strategies for adapting pre-trained language models for the encoding of textual attributes in TAGs, which is very common strategies. The technique contributions of this paper are unclear.

2. This paper only focuses on text-based graphs. The applications are limited since there are various non-text-based graphs in real-world applications, such as molecular graphs.

3. The experimental settings for evaluating graph contrastive learning are weak. In appendix, the authors state the settinng ". For Compt, Photo, and Hist datasets, we follow the few-shot learning setting (Garcia & Bruna, 2017) to randomly generate 300/30/1000 labeled nodes
per class for the train/validation/test sets.". Current graph contrastive learning methods commonly use 10%/10%/80% for evaluations[1]. I do not get the point of using a few-shot learning setting.

4. There lack comparisons with existing baselines. They propose different graph augmentation methods, and existing GCL methods mainly focus on proposing graph augmentation methods. The authors should compare more methods with existing GCL methods, such as GCA[1].



[1] Zhu, Yanqiao, et al. "Graph contrastive learning with adaptive augmentation." Proceedings of the Web Conference 2021. 2021.

**Details Of Ethics Concerns:**

same as Questions

---

### Official Review · Reviewer_s2Lm · 2023-11-01

**Soundness:** 3 good
**Presentation:** 2 fair
**Contribution:** 3 good
**Rating:** 5
**Confidence:** 4

**Summary:**

This work proposes LLM4GCL a pipeline for integrating LLM based augmentations. It introduces 3 prompts/pipelines for augmenting text features of a node using its features and neighborhood.
1) Structure-Aware summary: Uses node features and neighbors to revise the textual data of the central node
2) Indpendent Reasoning: Classifies node based on text attributes
3) Structure-Aware Reasoning: Classifies node based on text and structure
It also proposes a graph structure augmentation using an LLM, by giving an anchor node and candidate neighbors and having the LLM predict if they are connected or not.

They compare against a suite of different graph augmentations and different GCL techniques. The results show that ChatGAug (theirs) is significantly better on downstream tasks for PubMed than the baselines.

Their analysis on the 3 text feature augmentations show that they each are superior in different cases and it is task specific.

Appendix is missing and there are many references to the appendix. This needs to be added ASAP to be able to review the paper fully and understand the experimental setup. Will reassess after appendix is added.

**Strengths:**

Originality - This work is original and presents a novel strategy for augmenting text and graph structure features for graph contrastive learning
Quality - Seems high quality. Some datasets do not seem to have full results (only see ChatGAug results on PubMed). Maybe this is in appendix?
Clarity -  Needs appendix to be able to evaluate
Significance - This work represents what seems to be a potentially impactful strategy for graph augmentation for GCL. Has potential for both academic and industry impact.

**Weaknesses:**

I have some questions/concerns with some of the experimental set up. However, they may be explained in the appendix.

Main concern is that the graph reasoning using IDR and SDR appear to just be zero-shot predictions using ChatGPT? I'm not sure this is a fair experimental setup. Does this even count as graph augmentation or is it closer to ensembling? Would be helpful to see ChatGPT zero shot predictions on these tasks using IDR or SDR. In fact, would also be helpful to see few-shot ICL predictions as well.

I guess another question is are these augmented features only used in the contrastive unsupervised portion of the learning or at inference time as well? Either are okay but would have different implications.

The organization of the paper needs some improvement. Table 1 does not show any results for ChatGAug.

**Questions:**

Adjacency is R^{NxN} or {0,1}^{NxN} ?

Is ChatGPT making predictions about feature categories or classes being predicted? Appendix might explain

“Hence, defining a unified language model that exhibits strong generalization across various methods and datasets represents a promising avenue for future research.” Seems like bigger is better is the answer here. Much more powerful language models already exist.

Semi-supervised setting. Detail this more. Are some of the labels available as features during GCL? Or only during the few shot examples.

---

### Meta-Review · Area_Chair_VTYd · 2023-12-06

**Metareview:**

1. Summary: The paper introduces LLM4GCL, an approach to graph contrastive learning (GCL) on text-attributed graphs (TAGs; e.g., citation network). It explores two pipelines: LLM-as-GraphAugmentor, directly using LLMs to augment graphs at feature and structure levels, and LLM-as-TextEncoder, employing LLMs to encode textual attributes into embedding vectors.
2. Strengths: Reviewers acknowledge that this work is original and presents a novel strategy for augmenting text and graph structure features for graph contrastive learning. The proposed method is evaluated on six benchmark datasets, showcasing its promise in enhancing state-of-the-art GCL methods. The comparison with baseline techniques, including the development of ChatGAug, reveals significant improvements, particularly in downstream tasks for PubMed.
3. Weaknesses: Reviewers raise concerns about the experimental setup, particularly regarding graph reasoning with IDR and SDR; as a result, reviewers also point out further ablation studies that could have been done to compare to more relevant benchmarks.

**Justification For Why Not Higher Score:**

Reviewers were unanimously concerned about the presentation of the work and the soundness of the conclusion.

**Justification For Why Not Lower Score:**

n/a

---

### Decision · Program_Chairs · 2024-01-16

Reject